# FLUid management and InDividualized resuscitation in Sepsis (FLUIDS)—A Study protocol for a single-centre, open-label, randomized clinical trial

Sanne Ter Horst[1,2]*, Ewoud ter Avest[1], Jildou van Everdink[2], Matijs van Meurs[3], Tycho J. Olgers[2], Jaap Jan Vos[4], Kevin Damman[5], Jan C. ter Maaten[1], Hjalmar R. Bouma[2,6]

1 Department of Acute Care, University Medical Center Groningen, University of Groningen, Groningen, The Netherlands, 2 Department of Internal Medicine, University Medical Center Groningen, University of Groningen, Groningen, The Netherlands, 3 Department of Critical Care, University Medical Center Groningen, University of Groningen, Groningen, The Netherlands, 4 Department of Anaesthesiology, University Medical Center Groningen, University of Groningen, Groningen, The Netherlands, 5 Department of Cardiology, University Medical Center Groningen, University of Groningen, Groningen, The Netherlands, 6 Department of Clinical Pharmacy & Pharmacology, University Medical Center Groningen, University of Groningen, Groningen, The Netherlands

* s.ter.horst@umcg.nl

## Abstract

### Introduction

Early septic shock results from complex hemodynamic disturbances, including vasodilation, capillary leakage, and hypovolemia, requiring timely resuscitation to prevent organ failure. Initial fluid administration in the emergency department (ED) varies widely, influenced by clinician experience, patient comorbidities, and infection source. Fluid therapy is frequently not tailored to individual patient responsiveness and tolerance, highlighting the importance of personalized resuscitation strategies.

### Objective

This study evaluates the effectiveness of a personalized hemodynamic resuscitation strategy during the initial management of sepsis in the ED, using continuous non-invasive stroke volume index (SVI) measurements to guide intravenous (IV) fluid therapy and early vasopressor initiation.

### Method

This single-centre, open-label, randomized clinical trial at the Emergency Department of the University Medical Centre Groningen, the Netherlands, will enroll 188 adults with suspected sepsis requiring hemodynamic resuscitation, defined by a mean arterial pressure (MAP) <70 mmHg, systolic blood pressure (SBP) <90 mmHg, shock

**Data availability statement:** No datasets were generated or analysed during the current study. All relevant data from this study will be made available upon study completion.

**Funding:** The author(s) received no specific funding for this work.

**Competing interests:** The authors have declared that no competing interests exist.

index >0.9, or lactate concentration >4.0 mmol/L. Deferred consent will be obtained within 30 days of ED presentation. Patients will be randomized to the intervention group, in which fluids and vasopressors will be guided by changes in SVI ($\Delta$SVI) after IV fluid administration, or to the control group, where fluids and vasopressors will be administered at the discretion of the treating physician during the initial 3-hour resuscitation phase. The primary endpoint is the volume of IV fluids administered within the first three hours after enrollment. Secondary endpoints include cumulative fluid balance, timing and dose of vasopressors, incidence of organ failure, hospital length of stay, mortality and fluid-related adverse events, including venous congestion assessed by point-of-care ultrasound.

## Expected results

We anticipate that SVI-guided resuscitation will reduce IV fluid volume compared to standard care. This personalized approach, tailoring fluid and vasopressor administration to the individual's hemodynamic needs, has the potential to improve outcomes for patients with sepsis.

## Trial registration

ClinicalTrials.gov NCT07009665

---

## 1. Introduction

Septic shock is a consequence of complex and overlapping pathophysiological processes that impair circulatory function [1–3]. These include systemic hypovolemia and vasodilation. Hypovolemia can occur due to a reduced fluid intake or losses such as vomiting and diarrhea (absolute hypovolemia), as well as from increased capillary permeability leading to fluid leakage and redistribution (relative hypovolemia) [3,4]. Systemic vasodilation, mediated by inflammatory cytokines, nitric oxide, and other vasoactive mediators that cause a loss of vascular tone, represents the predominant mechanism [5]. This profound vasodilation can significantly reduce venous return and compromise cardiac output even in patients who are euvolemic [5,6]. Myocardial depression may further impair circulatory function in some patients. Altogether, these vascular changes result in hypoperfusion at cellular level, organ failure, and a significantly increased risk of mortality [3,7]. Early recognition and timely initiation of hemodynamic resuscitation are essential to stabilize blood pressure, restore tissue perfusion, and support vital organ function, thereby reducing the risk of progression to multi-organ failure and death.

Hemodynamic resuscitation in sepsis primarily involves fluid administration and vasopressor therapy [8]. A growing body of evidence indicates that a uniform one-size-fits-all resuscitation strategy, such as currently recommended by the Surviving Sepsis Campaign (SSC) guidelines, may not be beneficial to all patients, given the observed heterogeneity in sepsis etiology and presentations [8–11]. While fluid

administration may benefit fluid-responsive patients who are able to increase their stroke volume following a fluid bolus [12,13], others may have limited fluid tolerance due to cardiac dysfunction or increased capillary permeability [14–17], placing them at risk for interstitial fluid accumulation and venous congestion [18–20] (Fig 1).

Personalized hemodynamic resuscitation, using stroke volume (SV) to guide fluid therapy to an individual's cardiovascular response, has the potential to improve outcomes of patients with sepsis by reducing the risk of fluid overload, while at the same time facilitating early optimization of organ perfusion [21]. While SV-guided fluid resuscitation has shown to be of benefit in reducing fluid-related complications in critically ill intensive care unit (ICU) patients [22,23], its effectiveness during the initial resuscitation of patients with early septic shock already in the emergency department (ED) remains unclear [24–26].

This single-center, open-label randomized controlled trial will evaluate the effectiveness of a personalized fluid resuscitation protocol, by comparing volume resuscitation based on continuous non-invasive stroke volume index (SVI) measurements with standard care during the initial three-hour resuscitation phase in the ED. We hypothesize that this personalized resuscitation strategy will reduce the volume of intravenous (IV) fluids administered and may facilitate earlier initiation of vasopressors, thereby minimizing the risk of fluid overload and optimizing organ perfusion in patients with early septic shock.

## 2. Methods

### 2.1. Study design and setting

This open label randomized clinical trial will evaluate the effectiveness of a personalized hemodynamic resuscitation strategy based on non-invasive measurements of stroke volume index (SVI) as measured with the Baxter Starling SV monitor compared to standard care in adult patients with suspected sepsis for whom hemodynamic resuscitation is indicated in the emergency department (ED). Patients will be screened for eligibility in the ED of the University Medical Centre Groningen (UMCG) by the Acutelines research team between 9 am and 5 pm on weekdays throughout the year. Eligible patients will be enrolled using deferred consent (by proxy) to ensure timely intervention and minimize selection bias in the acute setting. The clinical trial will span 12 months and will include 188 patients, as determined by the sample size calculation

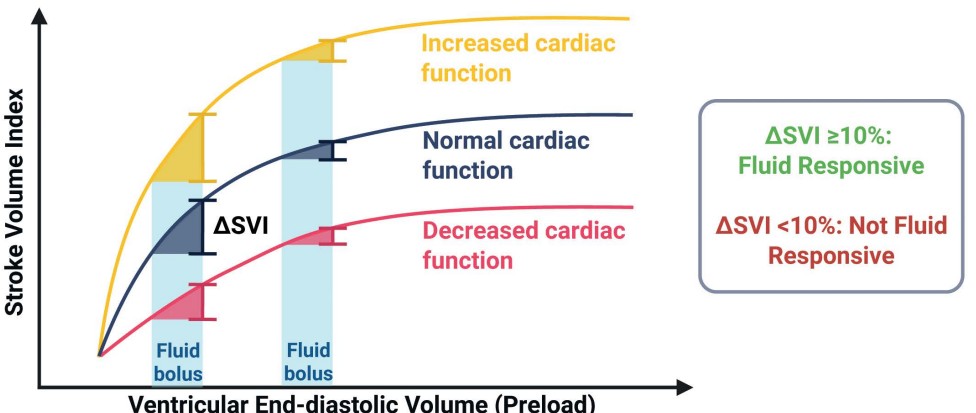

**Fig 1. Frank-Starling curve illustrating stroke volume response to fluid boluses and individual variation in cardiac contractility due to comorbidities and sepsis mechanisms.** The figure illustrates the relationship between preload, represented on the x-axis as ventricular end-diastolic volume, and stroke volume index on the y-axis, based on the Frank-Starling principle. Each curve represents a different level of cardiac contractility (decreased-normal-increased), which can vary between patients with sepsis. When a fluid bolus is given on the ascending portion of the curve (left side), it results in a significant increase in stroke volume index, indicating fluid responsiveness (ΔSVI ≥ 10%). However, on the plateau phase (right side), stroke volume does not appreciably rise following a bolus of IV fluids, demonstrating the patient is not fluid responsive. This conceptual model emphasizes that the effect of fluid administration depends both on the patient's current position on the curve and their underlying myocardial function. *Created in BioRender.*

(Section 3.1). Acutelines is a data-biobank at the UMCG, with a team of trained researchers able to screen/recruit patients for prospective clinical trials [27] (Fig 2). The trial is scheduled to start on September 1, 2025, and is estimated to run until September 1, 2026. We aim to present the results in the course of 2027.

## 2.2. Eligibility criteria

**2.2.1. Inclusion criteria.** Patients are eligible for inclusion if they are aged 18 years or older and present to the Emergency Department (ED) for Emergency Medicine, Internal Medicine (including all subspecialties), Gastro-Enterology or Pulmonology with suspected or confirmed sepsis requiring hemodynamic resuscitation. Sepsis is defined based on the physician's clinical judgment upon arrival at the ED, considering the presence of an acute phase response not attributable to a non-infectious cause (e.g., body temperature <36°C or >38°C, leukocyte count >12 × 10⁹/L, or C-reactive protein >50 mg/L), and/or symptoms suggestive of infection (e.g., productive cough, dyspnea, dysuria, pollakisuria, abdominal pain, erythema). This clinical suspicion must be accompanied by signs of hemodynamic instability requiring resuscitation, defined by one or more of the following criteria: mean arterial pressure (MAP) <70 mmHg, systolic blood pressure (SBP) <90 mmHg, shock index >0.9, or lactate level >4.0 mmol/L, based on initial measurements obtained either prehospital by emergency medical services (EMS) or upon ED arrival. The shock index is calculated as the ratio of heart rate to systolic blood pressure.

**2.2.2. Exclusion criteria.** Patients will be excluded when they received >1 liter IV fluid prior to randomization, when IV access cannot be gained, when non-invasive SVI and total peripheral resistance index (TPRI) measurements cannot be obtained reliably (e.g., due to intra-abdominal hypertension, ascites), or if the patient cannot be enrolled within 1 hour after ED presentation. Moreover, patients with a primary diagnosis of acute cerebral vascular event, acute coronary syndrome, acute pulmonary edema, status asthmaticus, major cardiac arrhythmia, drug overdose, burn or trauma-related injury, diabetic ketoacidosis, hyperosmolarity syndrome, or pancreatitis are also excluded. Other exclusion criteria include known aortic insufficiency, aortic abnormalities, or intraventricular heart defects (e.g., ventricular or atrial septal defects); advanced heart failure (NYHA class IV, on a waiting list for heart transplant, LVAD recipients, or chronic inotrope use), end-stage kidney disease (dialysis-dependent CKD stage 5 or eGFR < 15 mL/min/1.73 m²), and decompensated liver cirrhosis at ED admission (e.g., ascites, hepatic encephalopathy, or variceal bleeding). Additional exclusion include pregnancy, other causes of shock (e.g., hemorrhagic shock), anticipated surgery within 3 hours of presentation, or contraindications to vasopressor therapy (e.g., advanced care directive). Finally, patients transferred from another hospital after initiation of therapy or from another in-hospital setting will be excluded.

## 2.3. Randomization and blinding

Eligible patients will be randomized to either the intervention or control arm at the time of inclusion in the emergency department (Fig 2). Randomization will occur in a 1:1 allocation ratio using the Research Electronic Data Capture (RED-Cap, version 13.7.19) randomization module within REDCap tools hosted at the University Medical Center Groningen [28].

Due to the nature of the intervention, blinding of the treating physicians and nursing staff is not feasible. Investigators and treating physicians are aware of the assigned treatment strategy for each participant.

Patients can be excluded from the study if exclusion criteria are identified after randomization. All collected data from excluded participants will be deleted.

## 2.4. Interventions and study protocol

**2.4.1. Interventional group.** In the intervention group, during the first three hours of the ED visit, hemodynamic resuscitation will be guided by the response of SVI to fluid boluses as measured by non-invasive hemodynamic measurements using the Starling SV monitor (Starling Fluid Management Monitoring System, Baxter International Inc. [29]). The Starling SV monitor provides continuous, real-time measurements of SVI, cardiac index (CI), and TPRI. After

| TIMEPOINT** | Enrolment | Allocation | | | | | | |
|---|---|---|---|---|---|---|---|---|
| | *Triage* | *Baseline* | *Start resuscitation (0h)* | *End resuscitation (3h)* | *24h* | *48h* | *7d* | *30d* |
| **ENROLMENT:** | | | | | | | | |
| Eligibility screen | X | | | | | | | |
| Deferred consent | X | X | X | X | X | X | X | X |
| Randomisation | | X | | | | | | |
| Allocation | | X | | | | | | |
| **INTERVENTIONS:** | | | | | | | | |
| *Personalized SVI-guided hemodynamic resuscitation* | | | X | X | | | | |
| *Standard care hemodynamic resuscitation* | | | X | X | | | | |
| **ASSESSMENTS:** | | | | | | | | |
| *Demographic data (e.g. age, sex)* | | X | | | | | | |
| *Co-morbidity (pre-existent, developing)* | | X | | | X | X | X | X |
| *Medication use (e.g. cardiovascular, immunosuppresive)* | | | | | | | | |
| *Vital parameters (e.g. SBP, MAP, HR, CRT)* | | X | X | X | X | X | | |
| *Lab values (e.g. leukocytes, CRP, creatinine)* | | X | | | X | X | X* | |
| *Disease severity scores (e.g. SOFA, NEWS)* | | X | | X | X | X | X* | |
| *Volume of I.V. fluids (time, amount)* | X | X | X | X | X | X | | |
| *Organ support information (e.g. vasopressors, RRT, ventilation)* | X | X | X | X | X | X | X | X |
| *PoCUS assessments (e.g. IVC, VeXUS, BLUE)* | | | | X | X | | | |
| *(Electro)physiological waveform data (e.g. ECG, PPG)* | | X | X | X | | | | |
| *Hospital admission (incl. ICU admission)* | | X | X | X | X | X | X | X |
| *Mortality (in-hospital, out-of-hospital)* | | X | X | X | X | X | X | X |

**Fig 2. SPIRIT schedule of enrolment, interventions and assesments.** * Follow up of severity scores and lab values is up to 7 days or hospital ward discharge, whichever comes first; CRT: capillary refill time; ECG: electrocardiography; HR: heart rate; **I.**V.: intravenous; IVC: inferior vena cava; MAP: mean arterial pressure; NEWS: national early warning score; PoCUS: point-of-care ultrasound; PPG: photoplethysmography; RRT: renal replacement therapy; SBP: systolic blood pressure; SOFA: sequential organ failure assessment; SVI: stroke volume index; VExUS: venous excess ultrasound.

baseline measurements are performed, a first 500 mL fluid bolus in 10 minutes is administered. Thereafter, further fluid resuscitation is guided by the ΔSVI after each fluid bolus administration.

Specific protocolized recommendations are made for I.V. fluid and vasopressor administration based on the effect on SVI (fluid responsiveness = ΔSVI ≥10%) and the measured blood pressure, heart rate and SI aiming for a MAP ≥ 65, a SBP ≥ 90, and a SI ≤ 0.9 (see Fig 3). When vasopressor therapy is recommended according to the protocol, the dose will be adjusted based on vital signs and re-adjusted every 10 minutes based on hemodynamic measurements.

Rescue fluids can be given at any time and are defined as a protocolized 500 mL IV bolus over 10 minutes, administered if patients meet predefined criteria, including severe or refractory hypotension, elevated lactate, sustained tachycardia, or extreme hypovolemia (see Fig 3).

**2.4.2. Control group.** In the control group, IV fluid and vasopressor administration will be determined at the discretion of the physician.

**2.4.3. Data collection and follow up.** In both groups, vital parameters, including blood pressure, heart rate, and capillary refill time (CRT), will be recorded at ED arrival, immediately before and after the first fluid bolus, and at least

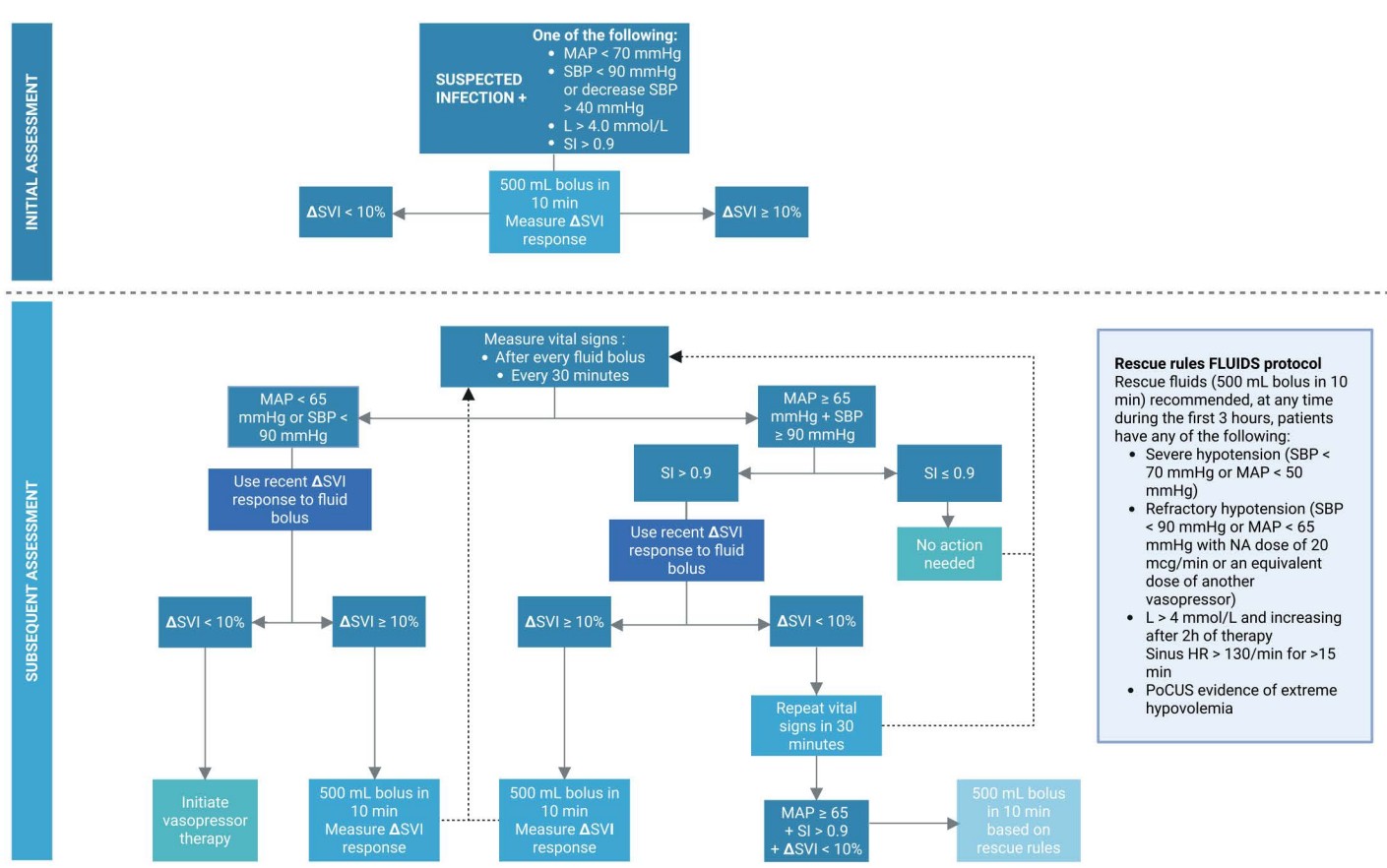

**Fig 3. Flowchart of the personalized protocol.** Adult patients with suspected or confirmed sepsis for whom hemodynamic resuscitation is indicated, will be included in the study. A dynamic assessment of fluid responsiveness using the Starling SV monitor will be performed at every clinical decision point, based on stroke volume index measurements and vital signs. Extreme hypovolemia is defined by an inferior vena cava diameter with a collapsibility index more than 75%, interpreted within the clinical context of suspected volume depletion. MAP: mean arterial pressure; SBP: systolic blood pressure; NE: noradrenaline; L: lactate; SI: shock index*; SVI: stroke volume index; IV: intravenous; *shock index is heart rate divided by systolic blood pressure. *Created in BioRender.*

twice per hour during the first three hours. Lactate will be measured at ED arrival and every two hours if initial lactate levels exceed 4 mmol/L. Capillary refill time (CRT) will be measured by applying firm pressure with a glass microscope slide to the ventral surface of the distal phalanx of the right index finger for 10 seconds [30].

Clinical characteristics, vital signs, laboratory tests, and treatment characteristics will be registered as part of the short-term secondary outcomes at 24 hours, 48 hours, 7 days and 30 days (see Table 1, Fig 4). Point-of-care ultrasound (PoCUS) will be performed twice within 48 hours after the emergency department resuscitation period, at 3 hours and 24 hours post-treatment, to evaluate fluid and congestion status using Inferior Vena Cava (IVC), Bedside Lung Ultrasound in Emergency (BLUE) protocol, and Venous Excess Ultrasound (VExUS) measurements [31,32].

## 2.5. Starling fluid management monitoring system

The Baxter's Starling Fluid Management Monitoring System is a portable, non-invasive cardiac output monitor (NICOM) that utilizes Bioreactance® technology [33]. The device operates using four electrode pads placed on the patient's chest. A pair of outer electrodes applies a low-frequency electrical current through the thorax, while the inner electrodes measure the resulting voltage. Changes in thoracic impedance, caused primarily by fluctuations in blood volume during the cardiac cycle, affect the voltage signal. By analyzing the phase shift and amplitude of the measured signal relative to the injected current, the system calculates stroke volume (SV). From the SV measurement, cardiac output (CO) is derived by multiplying SV by heart rate. Total peripheral resistance (TPR) is then calculated using mean arterial pressure (MAP) and CO. To account for individual differences in body size, all parameters are indexed to body surface area, which is calculated

**Table 1. Description and timeline of data collection.**

| Study periods | Screening[a] | ED visit[a] | | General ward/ ICU | | Follow up | |
|---|---|---|---|---|---|---|---|
| | | Baseline | | | | | |
| Hours/days | Triage | 0h | 3h | 24h | 48h | 7d | 30d |
| Deferred consent | | | | | | | x |
| Review of in-/exclusion criteria | x | | | | | | |
| Suspicion or confirmation of sepsis by treating physician | x | | | | | | |
| Demographic data (i.e., age, sex, ethnicity, length, weight, BMI) | | x | | | | | |
| Medication use (i.e., cardiovascular, immunosuppressives, diuretics) | | x | x | x | x | | |
| Co-morbidity (pre-existent, developing) | x | | x | x | x | x | x |
| Mortality (in-hospital, out-of-hospital) | | | x | x | x | x | x |
| Hospital admission (i.e., general ward, ICU) | | x | x | x | x | x | x |
| Disease severity scores (i.e., SIRS, [q]SOFA, NEWS) | x | x | x | x | x | x[b] | |
| Vital parameters (i.e., SBP, DBP, heart rate, respiratory rate, SpO2, temperature, GCS, CRT) | x | x | x | x | x | | |
| Lab values (i.e., leukocytes, thrombocytes, creatinine, ureum, bilirubin, CRP) | x | | | x | x | x[b] | |
| Extended lab values (i.e., pH, lactate, NT-proBNP) | x | | | x | | | |
| Organ support information (i.e., hemodynamic support, RRT, (non-) invasive mechanical ventilation, ECMO) | x | x | x | x | x | x | x |
| Volume of IV fluids (time, amount, type) | x | x | x | x | x | | |
| Vasopressor therapy (time, amount, type) | | x | x | x | x | | |
| (Electro)physiological waveform data: (e.g.,PPG, ECG) | | x | x | | | | |
| PoCUS data (e.g., IVC, VExUS, BLUE) | | | x | x | | | |
| Pre-tibial edema score | | x | x | x | | | |

[a]Screening and ED visit occur on the same day. [b] Follow up of severity scores and lab values is up to 7 days (or hospital discharge, whichever comes first);

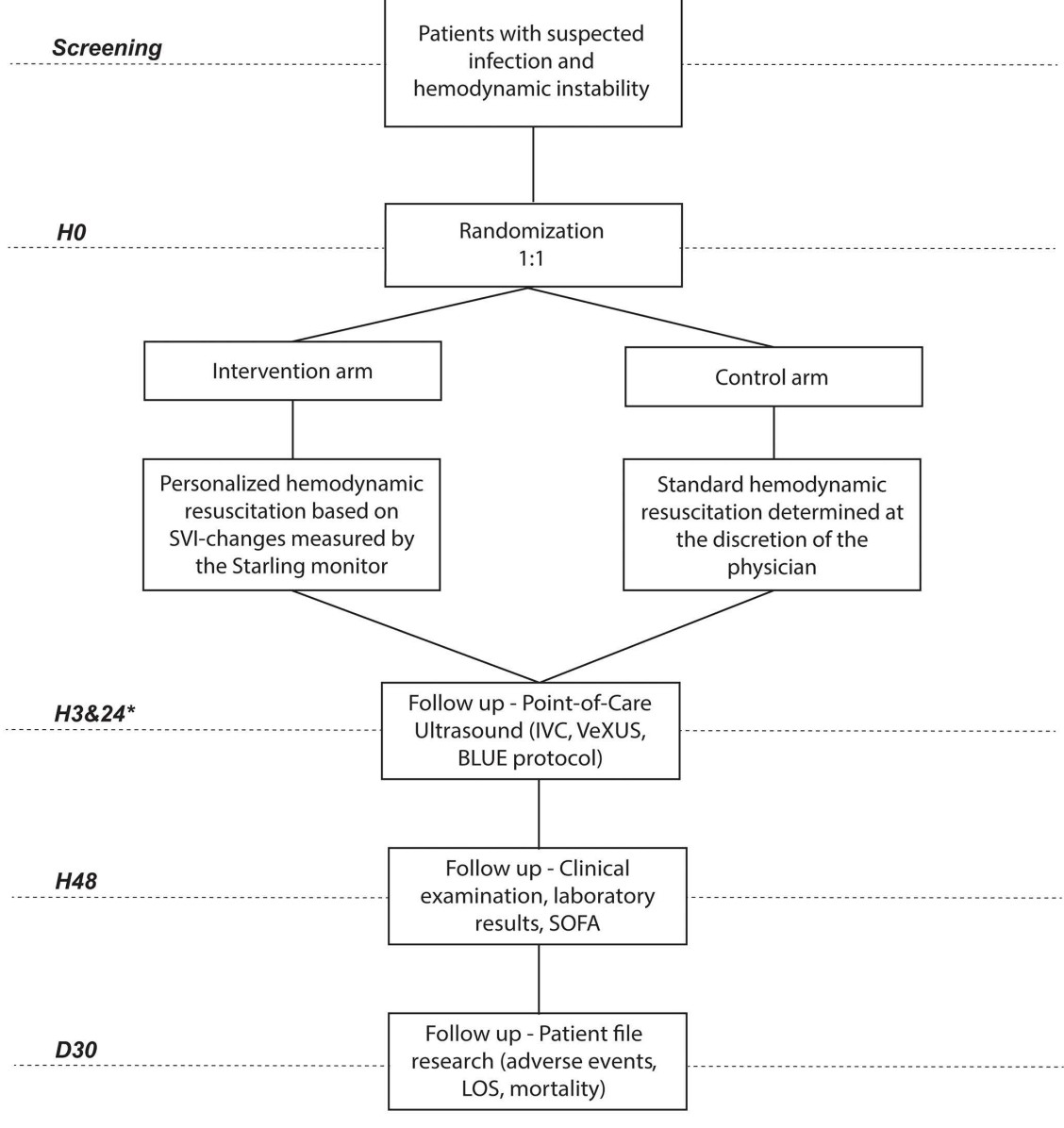

**Fig 4. Flowchart of patient enrollment, randomization, and follow-up.** This flow diagram illustrates the study design, including screening for eligibility, inclusion, and randomization into the intervention or control group. Point-of-Care Ultrasound assessments were conducted at 3 and 24 hours (*), with clinical follow-up (e.g., vitals parameters, laboratory values, fluid balance) at 48 hours post-randomization in both study arms. Clinical data were additionally collected through retrospective chart review up to 30 days following enrollment.

based on the patient's most recent weight and length. In this project, we use stroke volume index (SVI), cardiac index (CI), and total peripheral resistance index (TPRI) to assess hemodynamic dynamics in individual patients. The accuracy of this non-invasive technique has been evaluated in septic and critically ill patients [34,35]. While absolute agreement with thermodilution is limited (percentage errors often 48–67%) [36,37], bioreactance reliably detects relative changes in stroke volume. In elderly septic shock patients, a PLR-induced change in cardiac output measured by bioreactance predicted fluid responsiveness with 86% sensitivity and 79% specificity [38]. In mixed ICU cohorts, bioreactance showed

concordance rates of ~75–80% for tracking stroke volume changes compared with transpulmonary thermodilution [36,39], and accuracy improves when shorter averaging/refresh times are applied [36,40]. Thus, although absolute values are not interchangeable with thermodilution, the monitor provides clinically useful information on directional changes in stroke volume in septic patients.

## 2.6. Point of care ultrasound

Point-of-care ultrasound (PoCUS) is a rapid, noninvasive bedside tool increasingly used to assess hemodynamic status and guide fluid management in sepsis. Protocols including inferior vena cava (IVC) assessment, bedside lung ultrasound in emergency (BLUE), and the venous excess ultrasound (VExUS) score provide complementary information on intravascular volume, pulmonary congestion, and venous congestion [41–44]. PoCUS offers real-time qualitative and quantitative data, potentially detecting early physiologic changes not apparent on standard clinical examination. Importantly, fluid congestion can occur even in fluid-responsive patients [45], highlighting that fluid responsiveness and fluid tolerance are overlapping but distinct concepts.

To ensure consistent measurements, we implemented structured training, standardized acquisition protocols, image archiving, and blinded expert review [46,47], as detailed in **section 2.7** and **2.11**. Using this approach, we aim to monitor early signs of venous congestion and assess the safety of SVI-guided resuscitation, explore the hemodynamic interplay between fluid responsiveness and tolerance, and generate quantitative physiologic insights a to inform personalized resuscitation strategies and future studies.

## 2.7. Study procedures

**2.7.1. Starling SV monitor procedure.** Before measurements begin, the device will be inspected and set to auto-calibrate to ensure optimal performance and baseline stability. Subsequently, the four sensors will be attached to the patient's chest and connected to the Starling SV monitor. After one minute of calibration, CI, SVI, and TPRI readings are generated. If the monitor fails to deliver accurate readings, adjustments or replacements to the sensors will be made.

**2.7.2. Point-of-Care Ultrasound procedures.** PoCUS examinations will be performed according to standardized imaging protocols for assessment of fluid status. These protocols include assessing the IVC, VExUS, and the BLUE protocol. IVC assessment will include measurement of maximum and minimum diameters during the respiratory cycle to assess volume status, with calculation of the collapsibility index. The VExUS protocol includes Doppler assessments of the hepatic vein, portal vein, and intrarenal veins to evaluate venous congestion. The BLUE-protocol allows diagnosis of pulmonary edema by registering the appearance of B-lines. Operators will use appropriate ultrasound probes (e.g., curvilinear and phased array) to acquire images at predefined time points within the study. Detailed technical guidance for image acquisition, including probe selection and step-by-step instructions, is provided as supplementary material (see **S1 File**).

## 2.8. GDPR compliance and data safety

All data collection is compliant to the Dutch Personal Data Protection Act. All data will be pseudonymized, and a subject identification code list is used to link the data to the subject. The key to the code is safeguarded by the investigator(s). The code is a random number and will not contain any patient identifiable data. All data are stored in encrypted and password protected files on the study's research drive in the UMCG network. Research data will be stored for 15 years starting from the end of the study.

Data collected by the Starling SV monitor will be stored within the device under a pseudonymized study ID. Once the data collection period concludes, it will be downloaded and stored on a secure server of the UMCG for analysis. Additional data collected by the researcher will be entered directly into the REDcap database of the FLUIDS study [48]. A complete description and timeline of data collection are provided in Table 1 and Fig 4.

## 2.9. Ethics approval and informed consent

**2.9.1. Ethics approval.** This study was reviewed and approved by the Medical Ethics Committee of the Regional Review Board for Patient-related Research Leeuwarden (Regionale Toetsingscommissie Patiëntgebonden Onderzoek, RPTO) on 5 December 2024 (WMO study number NL84833.099.24). The research will be conducted in accordance with the principles outlined in the Declaration of Helsinki (latest version available at www.wma.net), the Medical Research Involving Human Subjects Act (WMO), and applicable guidelines, regulations, and laws. The protocol was registered in Clinicaltrials.gov, number **NCT07009665.**

**2.9.2. Informed consent.** Eligible patients will be enrolled upon arrival at the Emergency Department (ED). Due to the urgent nature of emergency care and the need for immediate treatment decisions, the Medical Ethics Committee has approved a deferred consent procedure [49,50]. The model consent form and participant information sheet (in Dutch) are provided as supplementary material (see S2 File).

Patients are initially enrolled in the study without prior consent due to the emergency setting. Whenever possible, the patient and/or their legally authorized representative (LAR) is informed about the study by a member of the research team. If neither the patient (or the LAR) verbally refuses participation, the patient is included in the study. Written informed consent will be sought from patients or their LAR within 30 days of ED admission. Study information will be provided as soon as possible after enrollment, either in the ED, ICU, or general ward. If the patient regains mental competence, their own written consent for continuation of participation must be obtained without undue delay. If the patient is discharged before consent can be obtained, they may be contacted up to three times by phone and/or mail. All contact moments and consent outcomes are documented in the REDCap screening log.

If a patient (or their representative) refuses consent when approached, they will be withdrawn from the study and all previously collected data will be deleted, except where retention is required by regulatory authorities for safety reporting. Participants who provide informed consent retain the right to withdraw from the study at any time. Data collected up to the point of withdrawal will be retained for research purposes.

## 2.10. Study endpoints

**2.10.1. Primary endpoint.** The primary endpoint is the volume of IV fluids administered within the first three hours after study enrollment.

**2.10.2. Secondary endpoints.** The key secondary outcomes of this study aim to assess the efficacy, safety, and cost-effectiveness of SVI-guided resuscitation compared with standard care. Efficacy endpoints include the cumulative amount of IV fluids and vasopressors administered in three hours, and the timing of vasopressors initiation. Additional efficacy endpoints include new-onset organ failure within 48 hours, defined by an increase in SOFA score of at least two points or the initiation of organ support. Hospital admission, length of hospital stay, ICU admission within seven days, and cumulative fluid resuscitation over seven days, assessed by total IV fluid volume and net fluid balance, are also included. Additionally, secondary measures focus on the time to achieve predefined hemodynamic resuscitation targets (MAP ≥ 70 mmHg, SBP ≥ 90 mmHg, lactate ≤ 4 mmol/L, ≥ 30% lactate decrease in those with baseline lactate > 2.5 mmol/L, and SI ≤ 0.9) and the proportion of patients meeting these endpoints within three hours.

Safety endpoints include the incidence of complications arising from either treatment strategy, such as fluid overload, which will be assessed for signs of congestion by point-of-care ultrasound at 3 hours and 24 hours using protocols such as IVC, VExUS, and BLUE, and by pre-tibial edema. Additional safety endpoints include the use of loop diuretics within seven days, the occurrence of congestive heart failure, respiratory insufficiency, or acute kidney injury, all monitored up to seven days, and the need for renal replacement therapy within 30 days. Major adverse cardiac events will be assessed up to 30 days, and mortality will be evaluated at 48 hours, during hospitalization, and at 30 days. At last, health economic assessments will focus on cost-effectiveness based on length of hospital stay, advanced care needs, intervention costs, and clinical health outcomes including readmissions and in-hospital/30-day mortality.

Exploratory mechanistic endpoints focus on understanding the underlying physiological processes and are intended to generate hypotheses for future research. These include (derived) hemodynamic parameters from the starling device (such as cardiac output, total peripheral resistance), lactate, measured within the first three hours after ED arrival, as well as photoplethysmography (PPG) and electrocardiography (ECG) waveform characteristics, which may offer insights into changes in cardiovascular function during the ED stay [51–53]. In addition, a set of blood samples will be collected and stored via the Acutelines data biobank infrastructure to enable future exploratory analyses of biomarkers associated with fluid responsiveness, venous congestion, and hemodynamic endotypes [27]. All endpoints are described in detail in the study protocol, which can be accessed on ClinicalTrials.gov for further reference.

## 2.11. Quality control and monitoring

Data quality will be maintained by a trained research team responsible for Starling SV monitor measurements, PoCUS assessments, and structured data collection. Competency must be demonstrated through supervised assessments prior to performing any measurements independently. Final evaluation and approval of individual competency will be carried out by experts in acute care with Entrustable Professional Activity (EPA) level 5 qualifications for POCUS. To ensure consistent PoCUS quality, expert reviewers with EPA level 4 in PoCUS will regularly assess image quality [54]. Ongoing training and structured evaluations will help maintain high standards for both PoCUS assessments and Starling SV monitor measurements throughout the study.

All study data will be securely captured via the Acutelines REDCap infrastructure, available 24/7 [27]. The FLUIDS investigators will provide ongoing support, with any missing or inconsistent data promptly identified and resolved in collaboration with site staff.

Monitoring will be conducted according to the guidelines of the *Nederlandse Federatie van Universitair Medische Centra* (NFU) and classified as a low-risk study. An on-site monitoring visit will be performed annually by a qualified study monitor from the UMCG. Adverse events are any undesirable experiences occurring during the study. All AEs reported spontaneously by the subject or observed by the investigator or his staff, will be recorded. Serious adverse events (SAEs) are defined as events potentially related to the study intervention and meet institutional criteria for seriousness (e.g., death, life-threatening event, hospitalization, persistent disability, or any other medically event). Events clearly attributable to the underlying disease, such as death from septic shock, are not considered SAEs. Reported SAEs will be documented in accordance with institutional and regulatory requirements to ensure participant safety throughout the study.

### 2.11.1. Missing data management.
As the amount of data to be collected for this study is substantial, we have implemented specific measures to mitigate the risk of missing data. Most importantly, we make use of dedicated research staff for this project, who can focus entirely on obtaining the measurement and follow-up data. Further, we use the ICT infrastructure developed by Acutelines that facilitates automated collection of data from the bed-side monitors [27]. Missing data will be reported transparently and managed in accordance with CONSORT guidelines for randomized controlled trials. For analyses, we will primarily apply data imputation techniques, such as multiple imputation, to minimize bias and maximize statistical power [55,56]. We acknowledge that multiple imputation assumes data are missing completely at random (MCAR) or missing at random (MAR) and recognize that if data are missing not at random (MNAR), this approach may not fully eliminate bias. In such cases, we will perform sensitivity analyses to assess the robustness of results. In instances where crucial variables, such as primary endpoints, are missing and cannot be reliably imputed, affected participants will be excluded from the corresponding analyses, with reasons and extent of exclusions clearly documented.

## 2.12. Patient and public involvement

SepsisNet, a collaborative network of clinicians, researchers, and healthcare professionals focused on sepsis care and patient outcomes, has contributed to the development of the study. Their patient-centered expertise ensures that the

perspectives, needs, and outcomes relevant to patients with sepsis are considered in the research process. The study protocol and patient information brochure has been reviewed by a former patient with sepsis.

## 3. Statistical methods

### 3.1. Sample size calculation

Sample size was calculated using R (version 4.3.1), based on data from a comparable study with similar methodology and endpoint definitions [22]. That reference study demonstrated a mean difference of 1.37 L in positive fluid balance at 72 hours between groups, with the intervention group showing 0.65 ± 2.85 L compared to 2.02 ± 3.44 L in the control group [22]. The effect size (Cohen's d = 0.45) was calculated from the reported means and standard deviations.

We conducted an a priori power analysis via a Monte Carlo simulation (50,000 replicates) assuming normal distributions for each arm with observed means and standard deviations. To accommodate unequal variances, a two-tailed Welch's t-test (α = 0.05) was applied in the simulated dataset. The simulation showed that 170 participants (85 per group) are needed to achieve 80% power to detect a statistically significant difference in the primary endpoint (IV fluid volume administered within the first three hours after study enrollment). Anticipating 10% drop-out due to loss to follow-up, study withdrawal, or protocol deviations, the total planned sample size is 188 participants (94 per group).

### 3.2. Statistical analysis plan

Primary and secondary outcomes will be analyzed based on intention-to-treat (ITT) for all randomized subjects. Descriptive statistics, such as mean, median, standard deviation, and confidence intervals, will be reported for continuous variables. Categorical variables will be summarized using frequencies and percentages.

In addition to the ITT analysis, we will conduct a per protocol (PP) sensitivity analysis and/or a modified intention-to-treat (mITT) analysis to assess the robustness of the primary results, regardless of the rate of protocol violations. Protocol violations are classified as minor (e.g., early study termination due to ICU transfer or discharge) or major. Major protocol violations are defined as failure to follow the assigned treatment protocol when clinically indicated, such as giving a fluid bolus when not indicated or not starting vasopressors when indicated. The per protocol (PP) analysis will include only participants who were strictly adherent to the study protocol without any major protocol violations. Minor protocol violations will be allowed if they do not impact the primary outcome. The modified intention-to-treat (mITT) analysis will include all eligible randomized participants except those with major protocol violations, while retaining participants with minor protocol violations. Participants who were found ineligible after randomization or from whom reliable Starling signals could not be obtained are excluded from all analyses as described above.

Statistical analysis will be conducted to compare outcomes between the treatment arms. The primary endpoint (volume of IV fluids administered within the first three hours) will be tested for normality using the Shapiro-Wilk test and analyzed using the appropriate statistical test. Continuous variables with a normal distribution will be analyzed using independent samples Student's t-tests, while non-normally distributed continuous variables will be analyzed using the Mann–Whitney U test. Categorical variables will be compared using the Chi-square test. A two-sided p-value of <0.05 will be considered statistically significant. To account for multiple comparisons among secondary outcomes, the false discovery rate (FDR) method will be applied where appropriate. Statistical analyses will be performed using R (version 4.3.1).

For binary outcomes, associations between treatment arms and clinical outcomes (e.g., fluid-related complications, new onset organ dysfunction, mortality) will be assessed using standard logistic regression analysis at pre-specified time points (e.g., in-hospital, 30-day outcomes). Time-to-event outcomes (e.g., time to vasopressor initiation, time to hemodynamic stabilization, time to ICU admission) will be evaluated using Cox proportional hazards regression. Additionally, longitudinal hemodynamic parameters (e.g., heart rate, blood pressure, stroke volume index) will be analyzed over time using mixed-effects models to characterize individual patient trajectories and assess how these trends relate to clinical

outcomes. All regression analyses will be adjusted for potential confounders, including age, sex, relevant comorbidities (e.g., heart failure, chronic obstructive pulmonary disease, chronic kidney diseases, and liver disease), and disease severity scores (e.g., NEWS, qSOFA).

Additionally, we will conduct a predefined subgroup analysis within the intervention arm based on fluid responsiveness. Fluid responders will be defined as patients with at least one documented increase in stroke volume index (SVI) of ≥10% following a fluid bolus, combined with achievement of hemodynamic stability within three hours without the need for vasopressor support. All other patients will be categorized as fluid non-responders. In parallel, unsupervised learning techniques (e.g., K-means clustering) will be applied to identify novel hemodynamic profiles using a combination of vital signs (e.g., blood pressure, heart rate, capillary refill time, fluid responsiveness), clinical characteristics, and laboratory parameters. We will investigate whether these predefined or data-driven profiles differ in terms of intravenous fluid volumes administered at three hours, other treatments received, and whether they are associated with clinical outcomes.

## 4. Strengths and limitations

The FLUIDS study aims to evaluate the benefit of a personalized hemodynamic resuscitation strategy, providing tailored interventions based on real-time fluid responsiveness measurements that reflect the dynamic state of the individual patient with sepsis in the ED. This approach, supported by advanced monitoring technologies, enables more accurate decision-making during the critical early resuscitation phase. By focusing on this early phase, when patients have received few or no therapies, the study captures the optimal window to assess hemodynamic status and guide resuscitation, unlike ICU studies, where patients have often already received substantial fluids. It also includes a broader ED population, from less severely ill patients to those developing refractory shock, capturing a diverse patient population in which optimal fluid strategies are often uncertain, particularly in those with comorbidities such as heart failure. Comprehensive data collection, including vital signs, capillary refill time, organ dysfunction, fluid balance, PoCUS assessments, physiological waveform data, and biomarker measurements, provides a multidimensional understanding of patient hemodynamics. The FLUIDS research team, part of Acutelines data-biobank, is well-trained, with expertise in both the Starling SV monitor and PoCUS, ensuring high-quality data collection and adherence to the study protocol. Moreover, the use of Acutelines ICT infrastructure supports the efficient and automated collection of data from the bed-side monitor. At last, the structured, personalized protocol in the intervention arm including systematic monitoring of fluid responsiveness and vital signs, along with predefined rescue rules, ensures consistent and reproducible treatment decisions, with the goal of providing optimal, individualized care for patients with sepsis in the ED.

The study has several limitations. First, the lack of blinding may introduce bias, as both the research and treatment teams are aware of treatment allocation, potentially influencing patient management and outcome assessments. Blinding the study protocol is not feasible, as both the researcher and treating physician must follow the protocol to ensure proper adherence. Moreover, awareness of fluid-related risks and benefits through trial information may lead physicians in the control (standard care) arm to adopt a more restrictive IV fluid strategy. Individual physician decision-making could further introduce variability in the control arm, which may affect the amount of IV fluids administered and should be considered when interpreting results. Second, conducting the study at a single center may limit the generalizability of the findings to other emergency departments with different patient populations or resources. Using IV fluid administered as the primary endpoint means the study is potentially not powered to detect differences in clinical outcomes, such as sepsis-associated organ dysfunction and mortality. However, this endpoint directly reflects the immediate effect of the intervention and is appropriate for a proof-of-principle study. If the results are promising, this single-center study will provide a foundation for future (international) multicenter studies.

In conclusion, the FLUIDS study combines a comprehensive design, a well-trained, experienced team, and structured personalized protocols using real-time fluid responsiveness monitoring in the emergency department. It aims to generate robust evidence on early, individualized fluid management, informing the design of future trials and ultimately improving FLUid management and InDividualized resuscitation in Sepsis.

## Supporting information

**S1 File. Point-of-Care Ultrasound Technical Performance in the FLUIDS Study.** Supplementary material detailing the technical protocol for performing point-of-care ultrasound assessments of the inferior vena cava (IVC), venous congestion (VExUS), and bedside lung ultrasound in emergency (BLUE) in the FLUIDS trial.
(DOCX)

**S2 File. The model consent form and participant information sheet (in Dutch).** Supplementary material containing the template participant information sheet and consent form in the study, provided in Dutch.
(PDF)

**S3 File. MeTC protocol FLUIDS clean copy.** Supplementary material containing the original study protocol, provided as copy in English.
(PDF)

**S4 File. SPIRIT checklist.** Supplementary material providing the completed SPIRIT (Standard Protocol Items: Recommendations for Interventional Trials) checklist for the FLUIDS study.
(DOC)

**S5 File. Fig Graphical abstract.** FLUid management and InDividualized resuscitation in Sepsis (FLUIDS). This illustration shows the study design of the FLUid management and InDividualized resuscitation in Sepsis (FLUIDS) trial, an open-label randomized controlled trial investigating personalized hemodynamic resuscitation in early septic shock at the emergency department. A total of 188 adult patients with suspected sepsis who require hemodynamic resuscitation are randomly assigned during the initial three-hour resuscitation phase to receive either fluid and vasopressor therapy guided by changes in stroke volume index (ΔSVI) or standard care based on the treating physician's discretion. The primary outcome is the volume of intravenous fluids administered within the first three hours after study enrollment. Secondary outcomes include the cumulative fluid balance, and the timing and dose of vasopressors, the incidence of organ failure, hospital length of stay, and venous congestion as assessed by point-of-care ultrasound. By continuously and non-invasively measuring fluid responsiveness, the study aims to offer a personalized approach to optimize fluid administration according to individual hemodynamic needs, which can improve organ perfusion and reduce the risk of fluid overload at the same time. Deferred consent will be obtained within 30 days after ED presentation. Created in BioRender.
(PDF)

## Acknowledgments

Special thanks go to Ilektra Koliaki, Jochem van Oosten, Ranek Laros, Lucia Colavolpe and Annet Eerens for their support in developing the Standard Operating Procedures and training framework. We would like to thank Prof. Dr. Jan Bakker for his expert input in the field of personalized hemodynamic resuscitation clinical trial development in sepsis. We would also like to thank dr. Nicole Erler for giving advice on the statistical plan. Finally, we extend our appreciation in advance to our colleagues – physicians, nurses, and researchers in the Emergency Department – for their cooperation and support during the implementation of this study.

**ClinicalTrials.gov identifier** NCT07009665

## Author contributions

**Conceptualization:** Sanne Ter Horst, Ewoud ter Avest, Jildou van Everdink, Matijs van Meurs, Tycho J. Olgers, Jaap Jan Vos, Kevin Damman, Jan C. ter Maaten, Hjalmar R. Bouma.

**Funding acquisition:** Sanne Ter Horst, Jan C. ter Maaten, Hjalmar R. Bouma.

**Investigation:** Sanne Ter Horst.

**Methodology:** Sanne Ter Horst, Ewoud ter Avest, Jildou van Everdink, Matijs van Meurs, Tycho J. Olgers, Jaap Jan Vos, Kevin Damman, Jan C. ter Maaten, Hjalmar R. Bouma.

**Project administration:** Sanne Ter Horst.

**Resources:** Sanne Ter Horst.

**Supervision:** Ewoud ter Avest, Jan C. ter Maaten, Hjalmar R. Bouma.

**Visualization:** Sanne Ter Horst, Jildou van Everdink.

**Writing – original draft:** Sanne Ter Horst, Ewoud ter Avest.

**Writing – review & editing:** Ewoud ter Avest, Jildou van Everdink, Matijs van Meurs, Tycho J. Olgers, Jaap Jan Vos, Kevin Damman, Jan C. ter Maaten, Hjalmar R. Bouma.

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
