## [Decision Letter · Decision Letter 0]

29 Aug 2025

Dear Dr. Ter Horst,

Thank you for submitting your manuscript to PLOS ONE. After careful consideration, we feel that it has merit but does not fully meet PLOS ONE’s publication criteria as it currently stands. Therefore, we invite you to submit a revised version of the manuscript that addresses the points raised during the review process.

We look forward to receiving your revised manuscript.

Kind regards,

Ronaldo Go, MD

Academic Editor

PLOS ONE

2. In the online submission form, you indicated that [No datasets were generated or analysed during the current study. All relevant data from this study will be made available upon study completion and upon reasonable request.].

Additional Editor Comments:

Reviewer #1:

Reviewer #2:

Reviewer #3:

Reviewer #4:

Reviewer #5:

Reviewers' comments:

Reviewer's Responses to Questions

**Comments to the Author**

1. Does the manuscript provide a valid rationale for the proposed study, with clearly identified and justified research questions?

Reviewer #1: Yes

Reviewer #2: Partly

Reviewer #3: Yes

Reviewer #4: Yes

Reviewer #5: Yes

2. Is the protocol technically sound and planned in a manner that will lead to a meaningful outcome and allow testing the stated hypotheses?

Reviewer #1: Yes

Reviewer #2: Partly

Reviewer #3: Yes

Reviewer #4: Yes

Reviewer #5: Yes

3. Is the methodology feasible and described in sufficient detail to allow the work to be replicable?

Reviewer #1: Yes

Reviewer #2: Yes

Reviewer #3: Yes

Reviewer #4: Yes

Reviewer #5: Yes

4. Have the authors described where all data underlying the findings will be made available when the study is complete?

Reviewer #1: Yes

Reviewer #2: No

Reviewer #3: Yes

Reviewer #4: Yes

Reviewer #5: No

5. Is the manuscript presented in an intelligible fashion and written in standard English?

Reviewer #1: Yes

Reviewer #2: Yes

Reviewer #3: Yes

Reviewer #4: Yes

Reviewer #5: Yes

You may also provide optional suggestions and comments to authors that they might find helpful in planning their study.

Reviewer #1: Thank you for the opportunity to review your manuscript which is a protocol for a randomized trial on fluid management and individualized resuscitation in sepsis. This trial will begin enrolling in the next couple weeks.

Introduction

- Well written; you could refer to the ESICM guideline on fluids which suggested an individualized approach to fluid resuscitation

Methods

- TPRI used in Exclusion criteria but spelt out later (minor revision)

- SAEs should be more prescriptive as opposed to what people spontaneously report

Overall, as you can see from the above, I only have minor revisions. This is a well written and thought out protocol.

Some things to consider for a larger version of this study (if you wish; just my opinion for your consideration as I understand that this study is starting soon):

- It may be advantageous to have a primary outcome that's a composite (i.e. mortality, AKI, need for RRT, fluid balance [dichotomized] etc)

- A TEE arm or including it as part of your protocol if good windows aren't available with TTE

- Consider measuring L sided pressures with TDI in protocol

- Should cirrhotic patients be excluded or be part of a subgroup? What about CKD/HD patients?

Reviewer #2: This is a protocol with ethics approval for a single site RCT comparing a non-invasive measure of cardiac output and fluid responsiveness to guide fluid management with routine clinician judgment in patients with a working diagniosis of sepsis

Recruitment will start soon (or has started).

Overall this is fairly well written, although I have a few comments which may help frame the paper/RCT a bit better.

There are a few key issues with the design and methodology.

1. Primary outcome is difference in fluid given in the first 3 hrs. This is really a feasibility and proof of concept outcome. If your intervention dictates a different volume given, then this will have an effect on your outcome. In other words – your intervention correlates with the outcome.

2. Furthermore, it would be good to speculate what a relevant difference is. The sample size was based on a study that showed a volume difference at 72 hours, not 3 hours. This means that even if your initial indivualised approach shows a effect, there may well be a catch up effect at 24 and 72 hours. Please explore and explain this. I suspect that you are heavily underpowered. In an ideal word I would think that the minimum possibly clinically important difference in volume in the first 3 hours would be 750-1000ml (based on Andrews et al https://pubmed.ncbi.nlm.nih.gov/28973227/ )

3. Cliniciians will be interested in duration of vaspossors, need for renal replacement, duration of ventilation and mortality

4. The complexity and accuracy of the intervention needs to be better spelled out. The paper states it is accurate compared to Swan Ganz, but I am not aware of any ICUs still using Swan Ganz cathers. All other papers do not seem to be in patients with sepsis.

5. Further, the accuracy (sens/spec vs which gold standard) of the POCUS measures (such as VEXUS and BLUE) need better justification. In my reading these protocols are still to be interpreted cautiously, especially with interrater reliability and absence of true gold standard. Fluid congestion is again a proxy outcome - possibly helpful to inform a future study.

The corollary of the above is that this really should be framed as a feasibility or pilot RCT to show that fluid volume separation can be achieved with this approach (defined as a volume not effect size), with the secondary clinical outcomes being measured which can inform a future multicentre RCT.

Further minor comments:

5. Abstract – The statement that clinicians use a fixed amount is not justified. It is quite clear from the literature that clinicians have large practice variation in hemodynamic support (https://pubmed.ncbi.nlm.nih.gov/32043315/ . This is probably based on clinician experience, setting, type of patient (respiratory sepsis may be treated more judiciously) and an already indvidualised approach (patient CCF vs renal failure will be approached differently anyway) e. These variable that may influence fluids administration should be considered as predefined subgroups (ED vs ICU, junior vs senior, source of sepsis, existing comorbidities)

6. Methods – are surgical causes of sepsis excluded?

7. SIRS criteria are mentioned but I wonder if it would be better to leave it at clinical judgment. SIRS was replaced for a reason (poor specificty)

8. I am also surprised to see the shock index as a measure. My understanding is that it performs poorly. At what time does the SI have to be >0.9. At any time?

9. 2.3 Randomisation

I am surprised that patients are randomised at triage. I assume you mean after meeting inclusion criteria? This could be at 20 min after triage, but patients may also develop sepsis 3 hrs into their ED stay, so only become eligible to be randomised then. Please clarify

10. More detail on accuracy (sens/spec vs which gold standard ) of the SV monitor needs to be in the body of the text. The references are now includings studies in dogs (Beagles) and patients with pulmonary hypertension, but no reference discusses sepsis. I think the reader needs to be convinced that this monitor is actually doing what it is supposed to and not a random number generator

Reviewer #3: The manuscript is generally well written; however, it can be further improved based on the comments below.

Line 24-32 Abstract: In the objectives section, the write-up from the introduction should be separated from the actual objective.

Line 33-47: A separate subtitle, ‘Method,’ should be included, with all relevant sections organized under this heading.

Line 242: Table 1: The symbol ** is missing in the table.

Line 298: Focusing only on hospital stay duration, advanced care needs, and intervention costs in health economic assessments is insufficient. It would be more comprehensive to also include parameters such as quality of life, health outcomes, and indirect costs.

Line 340: The multiple imputation (MI) method is applicable for most cases of missing data, especially if the missing data is missing completely at random (MCAR) or missing at random (MAR) and missingness is moderate (e.g., less than 30%). If the missing data is missing not at random (MNAR), MI may not fully correct the bias.

Line 358: In the G*Power sample size calculation, more details need to be specified, such as the test family, statistical test, type of power analysis, and whether a one- or two-tailed test was applied.

Line 366: In addition to ITT, conducting a per-protocol (PP) analysis as a sensitivity or secondary analysis is recommended to demonstrate the robustness of the results.

Line 370: Even if protocol violations are rare (<5%), it is still recommended to report both ITT and PP analyses.

Line 373: The definition of what constitutes major protocol deviations is to be stated.

Line 374: The sentence requires improvement e.g. Per-protocol (PP) analysis will include only subjects who strictly adhered to the study protocol and had no major protocol violations. Minor deviations will be permitted if the deviations do not impact the primary outcome/endpoint.

Line 389: The type of logistic regression should be specified (e.g., Standard Logistic Regression for a single endpoint at a fixed time, Repeated Measures Logistic Regression, or Mixed-Effects Logistic Regression). In addition, the time point(s) of analysis should be clearly stated.

Line 409: The statistical software used, including the publisher and version, is to be stated.

Line 436-445: The paragraph repeats certain words unnecessarily and could be refined. E.g, ‘In conclusion’, ‘Despite limitations such as lack of blinding and a single-center setting’.

Figure 4: The assessment time point is to be highlighted.

The format of references in the list is to follow the journal format.

Reviewer #4: Intro

Gives good background on the topic and clearly states the aim of the study

Methods

I would advise explaining where the 174 enrollees number came from. I see this is discussed in detail later in the paper, but I would suggest just stating this is from power calculation and refer to the appropriate section. Otherwise this number stands out awkwardly at this point in the paper.

I would advise defining what is meant by rescue fluids and whether that is a protocoled amount or based on clinical gestalt.

Questions— is POCUS only going to be used in the SVI-guided resuscitation group or in both the intervention and control groups? Also, what is the purpose of collecting POCUS data—is it purely data gathering or is there an intent to correlate results to NICOM data and associated interventions?

Note: there are two sections labeled 2.9—may want to correct the numbering or sub label one (i.e. 2.9a).

This Section otherwise seems complete and well thought out.

Statistical Methods

These seem appropriate.

Strengths/Limitations

The discussion on the importance of the study and acknowledging its limitations are appropriate and well-considered.

Reviewer #5: The FLUIDS protocol is a randomized clinical trial that evaluates the effectiveness of individualized fluid resuscitation in patients with sepsis. The trial uses continuous, noninvasive stroke volume index (SVI) monitoring in the emergency department. Given ongoing debates on optimal fluid management in sepsis, the research question is highly relevant. The manuscript adheres to SPIRIT recommendations and presents a clear rationale, making it generally well prepared.

I would like to congratulate the authors on their excellent work.

Informed consent

The deferred consent procedure is appropriate for the emergency context, but further detail is needed in the manuscript text (not only in the supplement) on how patients or legal representatives will be approached, and how data will be handled in cases of refusal.

End-points

The list of secondary outcomes is extensive and includes efficacy, safety, and exploratory endpoints (organ failure, congestion, biomarkers, mortality). Consider prioritising or categorizing secondary outcomes (key vs exploratory), to avoid issues with multiplicity and to focus the trial’s interpretation.

Statistical plan

The protocol includes ITT, per-protocol, and modified ITT analyses, which is appropriate. However, further clarification is needed on:

1) Adjustments for multiple comparisons given the large number of secondary outcomes.

2) The exact covariates included in adjusted regression models (age, sex, comorbidities are mentioned, but disease severity scores could also be considered).

Discussion and limitations

The discussion rightly notes the single-centre design and lack of blinding. It should also mention:

1) Possible operator-dependence of PoCUS and Starling measurements.

2) Restricted recruitment hours (9–17 on weekdays) which may introduce selection bias.

3) The challenge of generalisability to other ED settings with different patient populations and resources.

**Do you want your identity to be public for this peer review?** For information about this choice, including consent withdrawal, please see our Privacy Policy

Reviewer #1: No

Reviewer #2: **Yes: ** Gerben Keijzers

Reviewer #3: No

Reviewer #4: No

Reviewer #5: No

---

## [Author Response · Author response to Decision Letter 1]

28 Sep 2025

Reviewer #1:

Thank you for the opportunity to review your manuscript which is a protocol for a randomized trial on fluid management and individualized resuscitation in sepsis. This trial will begin enrolling in the next couple weeks.

Introduction

- Well written; you could refer to the ESICM guideline aon fluids which suggested an individualized approach to fluid resuscitation

We have added the recent ESICM guideline on fluids, which recommends an individualized approach to fluid resuscitation, as suggested.

Methods

- TPRI used in Exclusion criteria but spelt out laterTPRI (minor revision)

We have adjusted the text appropriately.

- SAEs should be more prescriptive as opposed to what people spontaneously report

We have revised the manuscript to clarify that serious adverse events (SAEs) are defined according to institutional criteria and are only reported if potentially related to the study intervention or if the relationship is uncertain. Events clearly attributable to the underlying disease (e.g., death from septic shock) are not considered SAEs. This provides a more prescriptive definition in line with regulatory standards while distinguishing SAEs from spontaneously reported adverse events.

Revised text:

“Adverse events are any undesirable experiences occurring during the study. All AEs reported spontaneously by the subject or observed by the investigator or his staff, will be recorded. Serious adverse events (SAEs) are defined as events potentially related to the study intervention and meet institutional criteria for seriousness (e.g., death, life-threatening event, hospitalization, persistent disability, or any other medically event). Events clearly attributable to the underlying disease, such as death from septic shock, are not considered SAEs. Reported SAEs will be documented in accordance with institutional and regulatory requirements to ensure participant safety throughout the study.”

Overall, as you can see from the above, I only have minor revisions. This is a well written and thought out protocol.

Some things to consider for a larger version of this study (if you wish; just my opinion for your consideration as I understand that this study is starting soon):

- It may be advantageous to have a primary outcome that's a composite (i.e. mortality, AKI, need for RRT, fluid balance [dichotomized] etc)

- A TEE arm or including it as part of your protocol if good windows aren't available with TTE

- Consider measuring L sided pressures with TDI in protocol

- Should cirrhotic patients be excluded or be part of a subgroup? What about CKD/HD patients?

Thank you for your suggestions. For a future (multi)center studies, it would indeed be interesting to include clinical effect endpoints. In the current proof-of-principle study, however, our primary focus is on the direct effects of personalized hemodynamic guidance on fluid administration, aiming to demonstrate that this approach can reduce total fluid volume and allow earlier initiation of vasopressors. Key secondary endpoints do include organ dysfunction, mortality, need for renal replacement therapy, and fluid balances.

Regarding TEE, in the emergency department setting we aim to use non-invasive tools, as TEE is relatively invasive for spontaneously breathing or non-sedated patients. TEE could be considered in ICU research, but it is not feasible for our current ED study. In future ED-ICU studies, its use could be considered. While measuring left-sided pressures with Tissue Doppler Imaging (TDI) can provide valuable hemodynamic information, it is challenging to implement in the emergency department setting due to the need for high-quality echocardiographic windows and trained operators, particularly in spontaneously breathing or unstable patients. TDI could be considered in future ED-ICU studies or as part of an optional sub-study.

As per your suggestion, we have added CKD/HD patients and patients with decompensated liver cirrhosis at arrival to the exclusion criteria. This update will be incorporated in a forthcoming amendment of our protocol.

Revised text:

“Patients will be excluded when they received >1 L IV fluid prior to randomization, when IV access cannot be gained, when non-invasive SVI and total peripheral resistance index (TPRI) measurements cannot be obtained reliably (e.g., due to intra-abdominal hypertension, ascites), or if the patient cannot be enrolled within 1 hour after ED presentation. Moreover, patients with a primary diagnosis of acute cerebral vascular event, acute coronary syndrome, acute pulmonary edema, status asthmaticus, major cardiac arrhythmia, drug overdose, burn or trauma-related injury, diabetic ketoacidosis, hyperosmolarity syndrome, or pancreatitis are also excluded. Other exclusion criteria include known aortic insufficiency, aortic abnormalities, or intraventricular heart defects (e.g., ventricular or atrial septal defects); advanced heart failure (NYHA class IV, on a waiting list for heart transplant, LVAD recipients, or chronic inotrope use), end-stage kidney disease (dialysis-dependent CKD stage 5 or eGFR <15 mL/min/1.73 m²), and decompensated liver cirrhosis at ED admission (e.g., ascites, hepatic encephalopathy, or variceal bleeding). Additional exclusion include pregnancy, other causes of shock (e.g., hemorrhagic shock), anticipated surgery within 3 hours of presentation, or contraindications to vasopressor therapy (e.g., advanced care directive). Finally, patients transferred from another hospital after initiation of therapy or from another in-hospital setting will be excluded.”

Reviewer #2:

This is a protocol with ethics approval for a single site RCT comparing a non-invasive measure of cardiac output and fluid responsiveness to guide fluid management with routine clinician judgment in patients with a working diagniosis of sepsis

Recruitment will start soon (or has started).

Overall this is fairly well written, although I have a few comments which may help frame the paper/RCT a bit better.

There are a few key issues with the design and methodology.

1. Primary outcome is difference in fluid given in the first 3 hrs. This is really a feasibility and proof of concept outcome. If your intervention dictates a different volume given, then this will have an effect on your outcome. In other words – your intervention correlates with the outcome.

The primary aim of our study is to demonstrate a proof-of-principle, assessing the direct impact of a personalized hemodynamic treatment approach. We acknowledge that the primary outcome (fluid volume administered in the first 3 hours) is influenced by the intervention itself. This is intentional: our intervention guides therapy based on real-time changes in stroke volume index rather than clinician judgment alone, allowing us to evaluate whether a personalized approach can more precisely tailor fluid therapy and avoid under- or over-resuscitation.

In this context, the correlation between intervention and outcome is not a limitation but a feature: reflecting the measurable effect of the intervention on clinical decision-making. Patient-centered endpoints such as organ failure, mortality, and hypervolemia are included as secondary outcomes, providing additional insight and informing sample size calculations for a larger multicenter follow-up trial.

Revised text:

“Using IV fluid administered as the primary endpoint means the study is potentially not powered to detect differences in clinical outcomes, such as sepsis-associated organ dysfunction and mortality. However, this endpoint directly reflects the immediate effect of the intervention and is appropriate for a proof-of-principle study. If the results are promising, this single-center study will provide a foundation for future (international) multicenter studies.”

2. Furthermore, it would be good to speculate what a relevant difference is. The sample size was based on a study that showed a volume difference at 72 hours, not 3 hours. This means that even if your initial indivualised approach shows a effect, there may well be a catch up effect at 24 and 72 hours. Please explore and explain this. I suspect that you are heavily underpowered. In an ideal word I would think that the minimum possibly clinically important difference in volume in the first 3 hours would be 750-1000ml (based on Andrews et al https://pubmed.ncbi.nlm.nih.gov/28973227/ )

Both Andrews et al. and Douglas et al. reported a mean difference in fluid administration of approximately 1.3–1.5 L, which helps define a clinically relevant effect. We chose Douglas et al. as the primary reference for our design because their SVI-guided methodology closely matches our intervention, even though their study assessed a later phase of resuscitation in the ICU after patients had already received ~2 L of fluids. Our study focuses on the early resuscitation phase in the emergency department, where the majority of fluids are typically administered within the first 3 hours. We recognize that a catch-up effect in fluid volumes may occur at later time points. Focusing on this early window allows us to evaluate whether individualized, stroke-volume–guided therapy meaningfully influences initial fluid administration.

Based on our updated calculations, accounting for unequal variance, 85 patients per arm (170 total), plus 10% dropout (188 total), is sufficient to assess the early impact of the intervention. By demonstrating feasibility and proof-of-principle, this study will provide important data to inform future studies that assess downstream clinical outcomes.

Revised text:

“Sample size was calculated using R (version 4.3.1), based on data from a comparable study with similar methodology and endpoint definitions (21). That reference study demonstrated a mean difference of 1.37 L in positive fluid balance at 72 hours between groups, with the intervention group showing 0.65 ± 2.85 L compared to 2.02 ± 3.44 L in the control group (21). The effect size (Cohen's d = 0.45) was calculated from the reported means and standard deviations.

We conducted an a priori power analysis via a Monte Carlo simulation (50,000 replicates) assuming normal distributions for each arm with observed means and standard deviations. To accommodate unequal variances, a two-tailed Welch’s t-test (α = 0.05) was applied in the simulated dataset. The simulation showed that 170 participants (85 per group) are needed to achieve 80% power to detect a statistically significant difference in the primary endpoint (IV fluid volume administered within the first three hours after study enrollment). Anticipating 10% drop-out due to loss to follow-up, study withdrawal, or protocol deviations, the total planned sample size is 188 participants (94 per group).”

3. Cliniciians will be interested in duration of vaspossors, need for renal replacement, duration of ventilation and mortalit

Vasopressor therapy, need for renal replacement therapy, invasive ventilation (both incidence and duration), and mortality are included as key secondary endpoints assessing efficacy and safety of the personalized resuscitation protocol compared to standard care.

4. The complexity and accuracy of the intervention needs to be better spelled out. The paper states it is accurate compared to Swan Ganz, but I am not aware of any ICUs still using Swan Ganz cathers. All other papers do not seem to be in patients with sepsis.

We have updated the information on the accuracy of the Starling device and included relevant references.

Revised text:

“The accuracy of this non-invasive technique has been evaluated in septic and critically ill patients. While absolute agreement with thermodilution is limited (percentage errors often 48–67%) (Galarza 2018, Shih 2016), bioreactance reliably detects relative changes in stroke volume. In elderly septic shock patients, a PLR-induced change in cardiac output measured by bioreactance predicted fluid responsiveness with 86% sensitivity and 79% specificity (Hu et al. 2017). In mixed ICU cohorts, bioreactance showed concordance rates of ~75–80% for tracking stroke volume changes compared with transpulmonary thermodilution (Zhu 2020, Galarza 2018), and accuracy improves when shorter averaging/refresh times are applied (Galarza 2018, Gavelli 2021). Thus, although absolute values are not interchangeable with thermodilution, the monitor provides clinically useful information on directional changes in stroke volume in septic patients.”

5. Further, the accuracy (sens/spec vs which gold standard) of the POCUS measures (such as VEXUS and BLUE) need better justification. In my reading these protocols are still to be interpreted cautiously, especially with interrater reliability and absence of true gold standard. Fluid congestion is again a proxy outcome - possibly helpful to inform a future study.

We agree with the reviewer that PoCUS protocols such as VeXUS and BLUE should be interpreted with caution given the absence of a golden standard and interrater variability. In our study, these measures are not used as a golden standard for validation of the SVI-guided resuscitation. Rather, they are applied to monitor signs of venous congestion and thereby assess the safety of the intervention. Importantly, congestion may occur even in fluid-responsive patients (Muñoz et al. 2024), underlining that fluid responsiveness and fluid tolerance are distinct. Our aim is to identify early signs of fluid overload during initial resuscitation, before organ dysfunction develops, and to clarify the relationship between responsiveness and tolerance to inform future personalized resuscitation strategies.

Added section to manuscript:

2.6 Point-of-Care Ultrasound

“Point-of-care ultrasound (POCUS) is a rapid, noninvasive bedside tool increasingly used to assess hemodynamic status and guide fluid management in sepsis. Protocols including inferior vena cava (IVC) assessment, bedside lung ultrasound in emergency (BLUE), and the venous excess ultrasound (VeXUS) score provide complementary information on intravascular volume, pulmonary congestion, and venous congestion (Argaiz, 2021; Arvig 2022; Ahn 2025; Klompmaker 2025). POCUS offers real-time qualitative and quantitative data, potentially detecting early physiologic changes not apparent on standard clinical examination. Importantly, fluid congestion can occur even in fluid-responsive patients (Munoz, 2024), highlighting that fluid responsiveness and fluid tolerance are overlapping but distinct concepts.

To ensure consistent measurements, we implemented structured training, standardized acquisition protocols, image archiving, and blinded expert review (Blanco 2016, Assavapokee 2024), as detailed in section 2.7 and 2.11. Using this approach, we aim to monitor early signs of venous congestion and assess the safety of SVI-guided resuscitation, explore the hemodynamic interplay between fluid responsiveness and tolerance, and generate quantitative physiologic insights a to inform personalized resuscitation strategies and future studies.”

The corollary of the above is that this really should be framed as a feasibility or pilot RCT to show that fluid volume separation can be achieved with this approach (defined as a volume not effect size), with the secondary clinical outcomes being measured which can inform a future multicentre RCT.

We have revised the discussion to emphasize the proof-of-principle nature of our study. We acknowledge the limitation of using fluid volume administered as the primary endpoint, as the study is likely not powered to detect differences in all sepsis-associated organ dysfunctions or mortality. Nevertheless, this proof-of-principle study is designed to generate valid evidence on early, individualized fluid management, demonstrating that separation in fluid volumes can be achieved. The data obtained will inform the design of future multicenter RCTs, ultimately aiming to improve personalized resuscitation and fluid management in sepsis.

Revised text:

“The FLUIDS study aims to evaluate the benefit

---

## [Decision Letter · Decision Letter 1]

25 Nov 2025

FLUid management and InDividualized resuscitation in Sepsis (FLUIDS) - a Study Protocol for a Single-Centre, Open-Label, Randomized Clinical Trial

PONE-D-25-39323R1

Dear Dr. Ter Horst,

We’re pleased to inform you that your manuscript has been judged scientifically suitable for publication and will be formally accepted for publication once it meets all outstanding technical requirements.

Kind regards,

Ronaldo Go, MD

Academic Editor

PLOS ONE

Additional Editor Comments (optional):

Reviewers' comments:

Reviewer's Responses to Questions

**Comments to the Author**

1. Does the manuscript provide a valid rationale for the proposed study, with clearly identified and justified research questions?

Reviewer #2: Yes

Reviewer #3: Yes

2. Is the protocol technically sound and planned in a manner that will lead to a meaningful outcome and allow testing the stated hypotheses?

Reviewer #2: Yes

Reviewer #3: Yes

3. Is the methodology feasible and described in sufficient detail to allow the work to be replicable?

Reviewer #2: Yes

Reviewer #3: Yes

4. Have the authors described where all data underlying the findings will be made available when the study is complete?

Reviewer #2: Yes

Reviewer #3: Yes

5. Is the manuscript presented in an intelligible fashion and written in standard English?

Reviewer #2: Yes

Reviewer #3: Yes

You may also provide optional suggestions and comments to authors that they might find helpful in planning their study.

Reviewer #2: Thanks for detailed and well considered responses.

I appreciate the justifications of comments which were reasonably rebutted

Reviewer #3: The authors have responded to the earlier comments, and the necessary revisions have been made. I have no further comments to add.

**Do you want your identity to be public for this peer review?** For information about this choice, including consent withdrawal, please see our Privacy Policy

Reviewer #2: **Yes: ** Gerben Keijzers

Reviewer #3: No

---

## [Editor Report · Acceptance letter]

PONE-D-25-39323R1

PLOS One

Dear Dr. Ter Horst,

I'm pleased to inform you that your manuscript has been deemed suitable for publication in PLOS One. Congratulations! Your manuscript is now being handed over to our production team.

Kind regards,

on behalf of

Dr. Ronaldo Go

Academic Editor

PLOS One